# Association between Antihyperlipidemic Agent Use and Age-Related Macular Degeneration in Patients with Hyperlipidemia: A Population-Based Retrospective Cohort Study

**DOI:** 10.3390/biomedicines11061508

**Published:** 2023-05-23

**Authors:** Chun-Hao Chen, Hsiu-Chen Lin, Hsiu-Li Lin, Joseph Jordan Keller, Li-Hsuan Wang

**Affiliations:** 1Division of Clinical Pharmacy, Department of Pharmacy, Taipei Veterans General Hospital, Taipei 112, Taiwan; 2Department of Pharmacy, School of Pharmacy, College of Pharmacy, Taipei Medical University, Taipei 110, Taiwan; 3Department of Pediatrics, School of Medicine, College of Medicine, Taipei Medical University, Taipei 110, Taiwan; 4Department of Clinical Pathology, Taipei Medical University Hospital, Taipei 110, Taiwan; 5Department of Neurology, General Cathay Hospital, Sijhih Branch, New Taipei City 221, Taiwan; 6Department of Psychiatry, Western Michigan University Homer Stryker M.D. School of Medicine, Kalamazoo, MI 49007, USA; 7Department of Pharmacy, Taipei Medical University Hospital, Taipei 110, Taiwan

**Keywords:** antihyperlipidemic agents, statin, fibrate, age-related macular degeneration, population-based retrospective cohort study

## Abstract

Several studies have indicated that lipoproteins might contribute to the pathogenesis of age-related macular degeneration (AMD). In this population-based retrospective cohort study, patients with hyperlipidemia were divided into two groups (study groups I and II) based on whether or not they were receiving antihyperlipidemic agents. The comparison group included patients without hyperlipidemia who were randomly selected and matched with study group II patients. A Cox proportional hazard model was used to evaluate the risk of AMD among the groups. Patients with hyperlipidemia receiving antihyperlipidemic agents (study group I, n = 15,482) had a significantly increased AMD risk (adjusted hazard ratio (HR) = 1.23, 95% confidence interval (CI) = 1.04–1.45) compared to those not receiving antihyperlipidemic agents (study group II, n = 15,482). However, with an increase in cumulative exposure, a reduced risk of AMD was observed in patients using a defined daily dose of more than 721, with an adjusted HR of 0.34 (95% CI = 0.22–0.53, *p* < 0.001). Additionally, the adjusted HR of AMD for study group II was 1.40 (95% CI = 1.20–1.63, *p* < 0.001) relative to the comparison group (n = 61,928). In conclusion, the study results indicated that patients with hyperlipidemia have a higher AMD risk than patients without hyperlipidemia. Furthermore, patients with hyperlipidemia who received antihyperlipidemic agents had a significantly increased AMD risk. However, a dose-dependent reduction in the risk of AMD was observed in patients with hyperlipidemia using statins or/and fibrates.

## 1. Introduction

Age-related macular degeneration (AMD) is the dominant cause of irreversible blindness in adults over the age of 50 years [1,2]. With a global prevalence of 170 million, it is estimated that at least 11 million individuals are diagnosed with AMD in the United States [3]. Aging is the primary risk factor. According to the National Eye Institute, the percentage of patients with AMD increased by 18%, from 1.75 million to 2.07 million from 2000 to 2010 in the United States. Moreover, the number of people with AMD is anticipated to increase to 5.44 million by 2050 [4]. White Americans account for the majority of AMD cases. A Taiwanese population-based prevalence survey in Shihpai reported that the prevalence rates of early AMD and late AMD were 1.9% and 9.2%, respectively [5]. Another Taiwanese study reported that the prevalence of AMD in Chinese individuals aged ≥ 65 years residing in Puzhi (in Taiwan) is similar to that in other large-scale international studies, irrespective of diagnosis with early or late AMD [6]. As a result of the phenomenon of a rapidly aging population in Taiwan, more attention should be paid to AMD prevention and treatment.

According to the Age-Related Eye Disease study (AREDS) [1], AMD is diagnosed based on the appearance of drusen, retinal pigmented epithelium (RPE) abnormalities, atrophy, and choroidal neovascularization. Particularly, the AMD stage can be determined based on the numbers, sizes, and distribution of drusen, which are composed of protein and lipids [7]. A previous study identified the association between hyperlipidemia and AMD [8]. According to Vassilev ZP et al.’s study [8], there is a small increased risk of AMD among patients with hyperlipidemia (OR = 1.08, 95% CI = 1.02–1.15). Recent evidence also suggested that high cholesterol levels are associated with several other eyes disease such as retinopathy, and have been also suggested in AMD [9]. To compare the frequency of co-medications used in patients with AMD and in controls, current use of lipid-lowering drugs was associated with an increase in the risk of AMD of 1.06 (1.01–1.12). Consequently, several studies have indicated that lipoproteins might contribute to the pathogenesis of AMD [10,11,12]. Statins and fibrates are the most commonly used antihyperlipidemic agents to treat hyperlipidemia. Recent studies have reported that statins may contribute to the pathogenesis of AMD through its anti-inflammatory function and reduction in atherosclerotic changes; however, the findings for this association remain controversial [13,14,15,16,17,18]. In Ma L et al.’s study [14], the authors found that statin use significantly reduced the risk of early AMD and had a significant protective effect for exudative AMD at the late stage. However, according to Gehlbach P et al.’s study [15], there is no evidence to conclude that statins have a role in preventing or delaying the onset or progression from currently available RCTs. Moreover, most of the relevant studies have not been conducted on Asian populations. Fibrates are antihyperlipidemic agents, and they have the potential to treat AMD by virtue of their antiangiogenic properties [19,20]. Nevertheless, studies of the relationship between fibrates and AMD are limited [19,20]. Therefore, we conducted this study to investigate the effects of antihyperlipidemic agents on the risk of AMD in Asian patients with hyperlipidemia in Taiwan.

## 2. Materials and Methods

### 2.1. Data Sources

We conducted a population-based retrospective cohort study from a database provided by Health and Welfare Data Science Center, Ministry of Health and Welfare, Taiwan. The National Health Insurance Research Database (NHIRD) contains medical records of 99.6% of more than 23 million Taiwanese (on, for example, their demographic characteristics, diagnosis, drug prescription, and examination results). We used the Longitudinal Health Insurance Database 2000 (LHID 2000), which contains 2,000,000 enrollees whose data were extracted randomly from the NHIRD and linked to medical records from 2000 to 2015. The distribution of enrollees does not significantly differ according to age, sex, or financial status. Random assignment can also be used to avoid selection bias.

Because all personal information was encrypted, this study was exempted from having to obtain written informed consent. This study was approved by the Joint Institutional Review Board of Taipei Medical University (reference number N201806005).

### 2.2. Study Population

In this cohort study, we divided the study population into three groups to investigate whether or not hyperlipidemia is associated with a risk of AMD. The enrollment period was between 1 January 2001, and 31 December 2010. Each group had a 5-year follow-up period.

#### 2.2.1. Study Group I

We selected patients with hyperlipidemia (International Classification of Diseases, Ninth Revision, Clinical Modification, ICD-9-CM code 272.X) who were aged over 50 years and had at least three hyperlipidemia diagnoses in their medical records spanning from 1 January 2001, to 31 December 2010. The date of antihyperlipidemic agent therapy initiation was considered the index date. The year in which enrollees were diagnosed with hyperlipidemia was defined as the index year. We excluded individuals who used antihyperlipidemic agents before hyperlipidemia diagnosis and who received antihyperlipidemic agents for less than 90 days within 365 days after first being administered with the agents. Furthermore, we excluded individuals who had AMD after receiving antihyperlipidemic agents within 1 year. People who had AMD before hyperlipidemia diagnosis were also excluded.

#### 2.2.2. Study Group II

This group included patients who had at least three hyperlipidemia diagnoses from 2001 to 2010. The date of the first hyperlipidemia diagnosis was the index date. From this group, we excluded people who had AMD diagnoses before hyperlipidemia. As opposed to the patients with a history of antihyperlipidemic agent use in study group I, study group II included patients who did not take any antihyperlipidemic agent. Each enrollee in study group II was matched with a study group I enrollee in a 1:1 ratio by age, sex, and index year.

#### 2.2.3. Comparison Group

Enrollees who did not have a hyperlipidemia diagnosis before 2010 and did not take any antihyperlipidemic agent before 2015 were included in the comparison group. Each enrollee in study group II was matched with four enrollees in the comparison group by age, sex, and index year. Patients in this group were assigned the same index date as that of the matched patients in study group II.

### 2.3. Primary Outcome Measurement

The primary outcome of this study was an individual event of two-time diagnoses of macular degeneration (ICD-9-CM code 362.50), nonexudative senile macular degeneration (ICD-9-CM code 362.51), exudative senile macular degeneration (ICD-9-CM code 362.52), and drusen (degenerative) (ICD-9-CM code 362.57), which may or may not be associated with antihyperlipidemic agent use (statins/fibrates).

### 2.4. Secondary Outcome Measurement

We also conducted a cumulative dosage analysis to determine the dose effect of statins and fibrates on the risk of AMD according to the defined daily dose (DDD) system recommended by the World Health Organization. DDD is the average maintenance dose per day for a drug when used for its main indication in an adult weighing 70 kg. The data on the total prescription dosage of statins and fibrates were extracted from the NHIRD. The DDDs for statins were as follows: atorvastatin, 20 mg/day; fluvastatin, 60 mg/day; lovastatin, 45 mg/day; pitavastatin, 2 mg/day; pravastatin, 30 mg/day; rosuvastatin, 10 mg/day; and simvastatin, 30 mg/day. Those for fibrates were as follows: bezafibrate, 600 mg/day; clofibrate, 2000 mg/day; etofibrate, 900 mg/day; fenofibrate, 200 mg/day; gemfibrozil, 1200 mg/day; simfibrate, 1125 mg/day.

### 2.5. Confounders

The confounders included comorbidities and medications adjusted prior to the index year. The comorbidities included as confounders were hypertension (ICD-9-CM codes 401–405), diabetes mellitus (ICD-9-CM code 250), cerebrovascular disease (ICD-9-CM codes 434–435), coronary heart disease (ICD-9-CM code 414), heart failure (ICD-9-CM code 428), atrial fibrillation (ICD-9-CM code 427.31), myocardial infarction (ICD-9-CM code 410), atherosclerosis (ICD-9-CM code 440.X), stroke (ICD-9-CM codes 430.X–437.X), peripheral vascular disease (ICD-9-CM codes 443.8X and 443.9), glaucoma (ICD-9-CM code 365.X), diabetic retinopathy (ICD-9-CM codes 362.01–362.02), obesity (ICD-9-CM codes 278.00–278.01), and tobacco use disorder/alcohol abuse (ICD-9-CM codes 305.0X, and 305.1). The medications recorded as confounders were aspirin, warfarin, hormone replacement therapy, nonsteroidal anti-inflammatory drugs (NSAIDs), antihypertensive agents, and cataract surgery. A multivariate analysis was performed to observe correlations between confounders simultaneously.

### 2.6. Statistical Analysis

The normality of the data is the first thing confirmed by the parametric test. We assessed the enrollees’ age and satisfied the normality in the end. A Student’s *t* test and Pearson’s chi-squared test were used to evaluate intergroup differences. Levene’s test also was performed simultaneously to assess the equality of variances for our study groups. A Cox proportional hazard model was used to estimate the risk of AMD among the groups. We estimated the 5-year AMD occurrence and analyzed the cumulative dosage using the Kaplan–Meier method and a log-rank test. Statistical significance was defined as a two-sided *p* value less than 0.05. All data analyses were performed using Statistic Analysis System (SAS 9.1 statistical package; SAS Institute, Cary, NC, USA).

## 3. Results

### Baseline Characteristics

Figure 1 illustrates the characteristics of groups I and II and the comparison group; 168,056 patients with records of three hyperlipidemia diagnoses from 2001 to 2010 were included. The study population comprised 15,482 patients who received antihyperlipidemic agents (study group I) and 15,482 patients who did not receive antihyperlipidemic agents (study group II) after exclusion and matching. The comparison group was composed of 61,928 enrollees without hyperlipidemia and who had not received antihyperlipidemic agents.

The baseline characteristics of the three groups are presented in Table 1. The mean age of the three groups was 61.74 ± 8.76 years, and men accounted for 44.03% of the study population. Patients with hyperlipidemia were more likely to develop comorbidities relative to individuals in the comparison group. Patients with hyperlipidemia also had a higher tendency of taking aspirin, warfarin, NSAID, or antihypertensive agents or receiving hormone replacement therapy. In multivariate analysis, no significant correlation was found between confounders and our outcomes after adjustment except for age and glaucoma (Appendix A, Table A1). During the 5-year follow-up period, there were 338 (2.18%) new AMD cases in study group I, 268 (1.73%) new cases in study group II, and 692 (1.12%) new cases in the comparison group. The data in Table 2 and Table 3 indicate that patients with hyperlipidemia who used antihyperlipidemic agents (statins and/or fibrates) had a significantly increased AMD risk (adjusted hazard ratio (HR) = 1.23, 95% confidence interval (CI) = 1.04–1.45) compared to those who did not use them. However, with an increase in cumulative exposure, AMD risk decreased in patients using more than 721 DDDs, with an adjusted HR of 0.34 (95% CI = 0.22–0.53, *p* < 0.001). Patients with hyperlipidemia had a higher significantly increased risk of AMD than patients without hyperlipidemia (adjusted HR = 1.40, 95% CI = 1.20–1.63) among those who did not use antihyperlipidemic agents. The subgroup analysis results shown in Table 3 indicate that patients receiving fibrates only and statins only exhibited an increased risk of AMD compared with patients in study group II (fibrate use only: adjusted HR = 1.76, 95% CI = 1.29–2.41; statin use only: adjusted HR = 1.23, 95% CI = 1.03–1.47). Patients receiving fibrates only exhibited a significantly higher risk than those receiving statins only (adjusted HR = 1.42, 95% CI = 1.04–1.96).

In the subanalysis of the cumulative dosage presented in Table 4, we found that patients receiving antihyperlipidemic agents of a ≤360 DDD had a greater AMD risk compared to patients in study group II (adjusted HR = 1.87, 95% CI = 1.57–2.22), but the results indicate a protective effect with ≥721 DDD (adjusted HR = 0.34, 95% CI = 0.22–0.53, *p* < 0.001). When the patients were divided into two groups based on the antihyperlipidemic agents used (fibrate use only vs. statin use only in the follow-up period), different trends were observed. Comparing the two results, the risk of AMD in the fibrate-only group was significantly higher than that in study group II, except for the 361–720 DDD group. In the statin-only group, only a ≤360 DDD was associated with a significantly increased AMD risk compared with study group II (adjusted HR = 1.49, 95% CI = 1.24–1.81). Although no statistically significant differences were observed in patients with a 361–720 DDD and ≥721 DDD, increasing protective effects were noted in patients using statins only. Moreover, the Kaplan–Meier analysis indicated that patients who had hyperlipidemia and were receiving antihyperlipidemic agents (study group I) had the highest cumulative incidence of AMD among the three groups (Figure 2). Patients with hyperlipidemia who did not use antihyperlipidemic agents (study group II) had a higher significantly increased risk of AMD than patients without hyperlipidemia who did not use antihyperlipidemic agents (comparison group) did. The log-rank test of these three groups revealed significant differences (*p* < 0.0001).

## 4. Discussion

Basal laminar deposits, the accumulation of lipoprotein particles, are important histopathological markers of AMD [21]. Several studies have demonstrated a potential correlation between AMD and atherosclerosis and have proposed possible pathogenesis [16,22]. Additionally, a cohort study indicated that high total cholesterol levels in early middle age may have a role in the initial development of AMD [23]. Therefore, we presume that hyperlipidemia is a risk factor of AMD and this hypothesis has been confirmed in our study results. The results indicated that patients who had hyperlipidemia and who did not receive antihyperlipidemic agents had a significantly higher risk of AMD relative to patients in the comparison group.

Regarding our primary outcome, the results showed that the use of statins and fibrates increased the risk of AMD. Our study finding regarding statin use is consistent with that of VanderBeek’s study. The study indicated that 1 year of statin use increased the risk of exudative AMD [24]. With regard to the results of the cumulative dosage analysis, we surprisingly found that patients receiving antihyperlipidemic agents of a ≤360 DDD had a greater AMD risk compared to patients in study group II, but the results indicated a protective effect at a higher DDD (≥721). In addition, in the subanalysis of the cumulative dosage of fibrate use only and statin use only groups, patients receiving fibrates only showed a significantly higher risk than those receiving statins only. A possible explanation for this is that increased HDL and decreased triglycerides are possible risk factors. Some studies conducted over the past two decades have provided crucial evidence of the relation between high-density lipoprotein (HDL) and AMD [25,26,27,28]. From the European Eye Epidemiology Consortium and EYE-RISK Consortium, a similar conclusion was found in a 2019 study, which showed that HDL was associated with an increased risk of AMD, whereas triglycerides were associated with a decreased risk [29]. Fibrates result in a substantial decrease in plasma triglycerides and have more prominent effects on HDL elevation than statins do [30,31,32]. Therefore, these factors may explain why patients receiving fibrates only showed a significantly higher risk of AMD. However, one study found that fenofibric acid has potent effects on ocular neovascularization in animal models [20]. This result may be explained by the fact that fenofibric acid suppresses vascular leakage and inhibits inflammation through PPARα activation. This conclusion is in conflict with our results; thus, further research should be undertaken to investigate the relation between fibrates and AMD.

A multicenter open-label prospective clinical pilot study reported that during a 12-month follow up, high-dose atorvastatin caused drusen to deposit regression, which was associated with vision gain and prevented conversion to neovascular AMD [33]. The result is likely to be related to the ability of statins to prevent lipid accumulation, reduce oxidative stress, and modulate ApoB100 secretion in human RPE cells. Regarding our secondary outcome, in the statin-only group, the findings were inconsistent for the low-dosage group (≤360 DDD) and the medium- and high-dosage groups (361–720 DDD and ≥721 DDD, respectively). As shown in Table 3, the use of statins only increased the risk of AMD, and a significant difference was observed in the low-dosage group (≤360 DDD), especially compared with study group II. These results may be due to HDL elevation, as mentioned earlier. There was a trend of a protective effect of statin in the other two groups (361–720 DDD and ≥721 DDD), although the difference was not significant. These nonsignificant results are likely due to the limited number of AMD patients who had high-dosage statin use.

Our study has several strengths. First, to the best of our knowledge, this is the first study to explore the effect of antihyperlipidemic agents, namely statins and fibrates, on AMD risk in East Asian patients. Furthermore, it is a large population-based cohort study that used data from the NHIRD, which contains the medical records of 99.6% of the Taiwanese population, reducing selection and recall bias. Second, our inclusion and exclusion criteria are rigorous, which allowed us to reduce classification bias. We only included patients aged over 50 years with at least three new hyperlipidemia diagnoses, and we ensured that patients had sufficient antihyperlipidemic agent exposure (≥90 days) for a sufficient period. In addition, we excluded patients who had AMD diagnoses before receiving antihyperlipidemic agents or who had AMD after receiving antihyperlipidemic agents within 1 year. Third, we are the first to analyze the DDD cumulative dosage to evaluate the relationship between antihyperlipidemic agents and AMD. Fourth, we considered hyperlipidemia in our analysis to prevent indication bias from affecting the results. Fifth, patients with AMD were defined as those with at least two diagnosis records to avoid coding inaccuracy. Finally, we adjusted for numerous potential confounders to decrease interference from other factors.

Nevertheless, limitations still exist. Patient information, including laboratory data, education/financial level, dietary habits, smoking, gene type, and disease severity, is not available in the NHIRD. These factors may have affected the outcomes because we could not adjust for confounders. Second, we did not determine subclasses of statins and fibrates in the time-to-event analysis. They may have beneficial or harmful effects causing AMD according to specific classes. Third, we could not collect data on patient compliance to and the side effects of the antihyperlipidemic agents. The safety to efficacy ratio still remains unknown. Although we could not collect data on patient compliance to antihyperlipidemic agents, all patients had a long-term use of statins and/or fibrates for hyperlipidemia treatment and did not use them accidentally. The cause–consequence relationship between antihyperlipidemic agents and the risk of AMD was considerably increased.

## 5. Conclusions

In conclusion, antihyperlipidemic agents have demonstrated protective effects against coronary heart disease. However, statins and/or fibrates may increase the risk of AMD in patients with hyperlipidemia with low-dosage usage (≤360 DDD). Moreover, a dose-dependent reduction in the risk of AMD was observed in patients with hyperlipidemia; a protective effect was found with the use of a ≥720 DDD upon cumulative dosage analysis. Meanwhile, hyperlipidemia is one of the risk factors in AMD which can be supported by our study. The underlying mechanisms remain unclear, and further studies are needed to confirm these results.

## Figures and Tables

**Figure 1 biomedicines-11-01508-f001:**
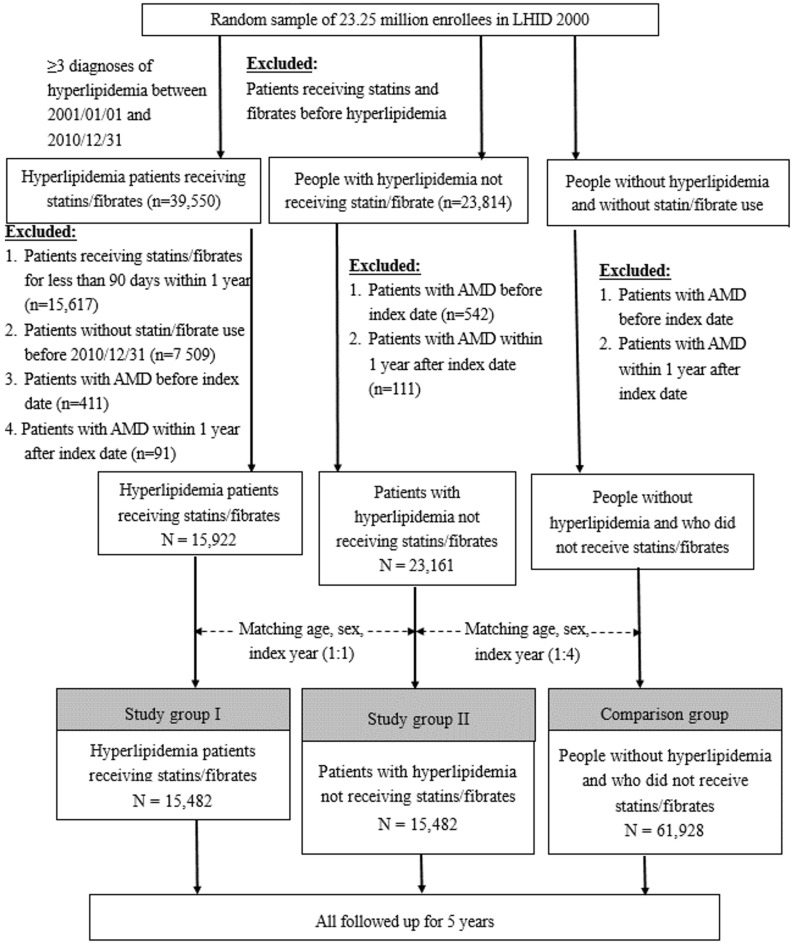
Flowchart of study design. LHID, Longitudinal Health Insurance Database; AMD, age-related macular degeneration; y/o, years old.

**Figure 2 biomedicines-11-01508-f002:**
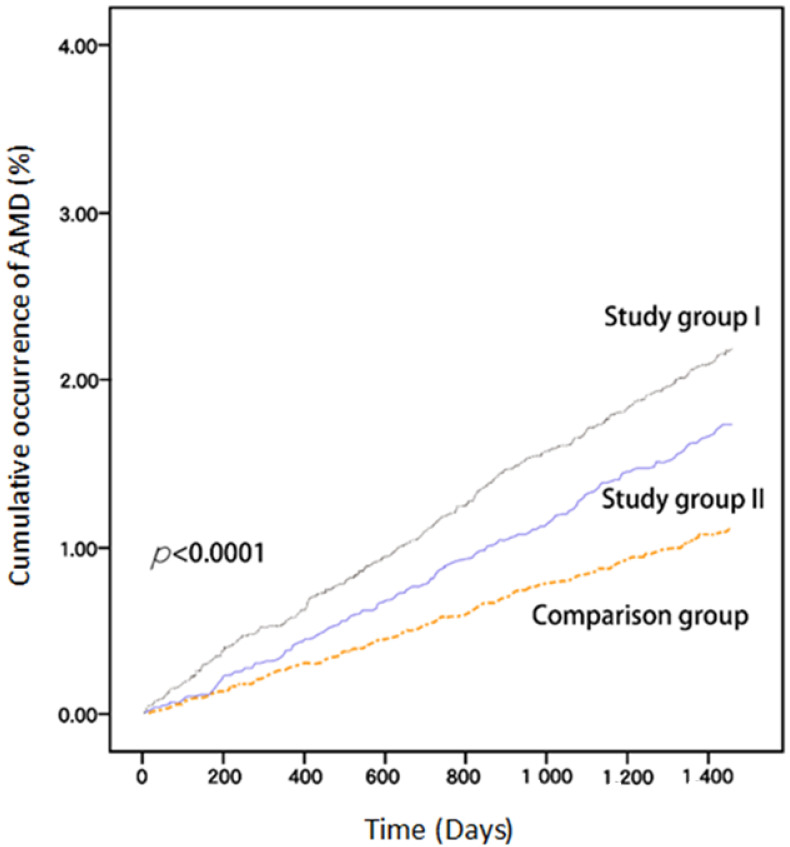
Kaplan–Meier curve of cumulative occurrence of AMD among the three groups: study group I, hyperlipidemia with antihyperlipidemia agents; study group II, hyperlipidemia without antihyperlipidemia agents; comparison group, nonhyperlipidemia without antihyperlipidemia agents. *p* < 0.0001 in log-rank test. AMD, age-related macular degeneration.

**Table 1 biomedicines-11-01508-t001:** Baseline characteristics of study and comparison groups.

Variables	Patients with Hyperlipidemia (2001–2010)			*p* Value ^a^	*p* Value ^b^
	Receive Statins/Fibrates (Study Group I)	Not Receive Statins/Fibrates (Study Group II)	Comparison Group		
	N = 15,482	N = 15,482	N = 61,928		
Age, years (mean ± SD)	61.74 ± 8.76	61.74 ± 8.76	61.74 ± 8.76	0.9928	0.9908
Sex/male (n, %)	6817 (44.03)	6817 (44.03)	27,268 (44.03)	1	1
Hypertension	11,586 (74.84)	9372 (60.53)	20,342 (32.85)	<0.0001	<0.0001
Diabetes mellitus	6662 (43.03)	5451 (35.21)	6220 (10.04)	<0.0001	<0.0001
Cerebrovascular disease	1989 (12.85)	1368 (8.84)	3433 (5.54)	<0.0001	<0.0001
Coronary artery disease	4902 (31.66)	3792 (24.49)	7505 (12.12)	<0.0001	<0.0001
Heart failure	1330 (8.59)	964 (6.23)	2707 (4.37)	<0.0001	<0.0001
Atrial fibrillation	291 (1.88)	246 (1.59)	734 (1.19)	0.0501	<0.0001
Myocardial infarction	270 (1.74)	110 (0.71)	281 (0.45)	<0.0001	<0.0001
Atherosclerosis	688 (4.44)	584 (3.77)	993 (1.60)	0.0029	<0.0001
Stroke	3219 (20.79)	2415 (15.60)	6100 (9.85)	<0.0001	<0.0001
Peripheral vascular disease	750 (4.84)	623 (4.02)	1370 (2.21)	0.0005	<0.0001
Glaucoma	1083 (7.0)	963 (6.22)	2582 (4.17)	0.006	<0.0001
Diabetic retinopathy	794 (5.13)	433 (2.80)	534 (0.86)	<0.0001	<0.0001
Cataract surgery	1421 (9.28)	1021 (6.59)	3485 (5.63)	<0.0001	<0.0001
Obesity	172 (1.11)	149 (0.96)	90 (0.15)	0.1969	<0.0001
Tobacco use disorder, alcohol abuse	243 (1.57)	227 (1.47)	669 (1.08)	0.4571	<0.0001
Aspirin	7352 (47.49)	5305 (34.27)	12,107 (19.55)	<0.0001	<0.0001
Warfarin	7206 (46.54)	5175 (33.43)	11,804 (19.06)	<0.0001	<0.0001
Hormone replacement therapy	701 (4.53)	734 (4.74)	1731 (2.80)	0.3724	<0.0001
NSAID	14,766 (95.38)	14,661 (94.70)	56,485 (91.21)	0.006	<0.0001
Antihypertensive agents	5895 (38.08)	5116 (33.04)	15,404 (24.87)	<0.0001	<0.0001

SD, standard deviation; NSAID, nonsteroidal anti-inflammatory drug. *p* value ^a^: study group I (patients with hyperlipidemia with the use of statins and/or fibrates) vs. study group II (patients with hyperlipidemia without the use of statins and fibrates). *p* value ^b^: study group II vs. comparison group (patients without hyperlipidemia without the use of statins and fibrates).

**Table 2 biomedicines-11-01508-t002:** Age-related macular degeneration risk of three groups.

Results	Patients with Hyperlipidemia		
	Receive Statins/Fibrates (Study Group I)	Not Receive Statins/Fibrates (Study Group II)	Comparison Group
	N = 15,482	N = 15,482	N = 61,928
AMD cases (n, %)	338 (2.18)	268 (1.73)	692 (1.12)
Crude HR (95% CI)	1.27 (1.08–1.48) **	1	-
Adjusted HR (95% CI)	1.23 (1.04–1.45) *	1	-
Crude HR (95% CI)	-	1.56 (1.35–1.79) ***	1
Adjusted HR (95% CI)	-	1.40 (1.20–1.63) ***	1

***: *p* < 0.001; **: *p* < 0.01; *: *p* < 0.05; HR, hazard ratio; AMD, age-related macular degeneration; study group I, patients with hyperlipidemia with the use of statins and/or fibrates; study group II, patients with hyperlipidemia without the use of statins and fibrates; comparison group, patients without hyperlipidemia without the use of statins and fibrates. HRs adjusted for age, gender, hypertension, diabetes mellitus, cerebrovascular disease, coronary artery disease, heart failure, atrial fibrillation, myocardial infarction, atherosclerosis, stroke, peripheral vascular disease, glaucoma, diabetic retinopathy, obesity, tobacco use disorder and alcohol abuse, cataract surgery, aspirin, warfarin, hormone replacement therapy, nonsteroidal anti-inflammatory drug, and antihypertensive agents.

**Table 3 biomedicines-11-01508-t003:** Age-related macular degeneration risk among fibrate-only group, statin-only group, and study group II.

Results	Receive Statins/Fibrates (Study Group I)		Not Receive Statins/Fibrates (Study Group II)
	Fibrates Only	Statins Only	
	N = 1492	N = 10,364	N = 15,482
AMD cases (n, %)	47 (3.15)	228 (2.20)	268 (1.73)
Crude HR (95% CI)	1.84 (1.35–2.51) **	1.28 (1.07–1.52) **	1
Adjusted HR (95% CI)	1.76 (1.29–2.41) **	1.23 (1.03–1.47) **	1
Crude HR (95% CI)	1.44 (1.05–1.97) *	1	-
Adjusted HR (95% CI)	1.42 (1.04–1.96) *	1	-

**: *p* < 0.01; *: *p* < 0.05; HR, hazard ratio; AMD, age-related macular degeneration; study group II, patients with hyperlipidemia without the use of statins and fibrates. HRs adjusted for age, gender, hypertension, diabetes mellitus, cerebrovascular disease, coronary artery disease, heart failure, atrial fibrillation, myocardial infarction, atherosclerosis, stroke, peripheral vascular disease, glaucoma, diabetic retinopathy, obesity, tobacco use disorder & alcohol abuse, cataract surgery, aspirin, warfarin, hormone replacement therapy, nonsteroidal anti-inflammatory drug, and antihypertensive agents.

**Table 4 biomedicines-11-01508-t004:** Effect of antihyperlipidemic agent exposure on age-related macular degeneration risk.

Study Group I	≤360 DDD	361–720 DDD	≥721 DDD	Study Group II
	N = 7678	N = 3768	N = 4036	N = 15,482
AMD cases (n, %)	266 (3.46)	50 (1.33)	22 (0.05)	268 (1.73)
Crude HR (95% CI)	2.02 (1.71–2.40) ***	0.76 (0.57–1.03)	0.31 (0.20–0.48) ***	1
Adjusted HR (95% CI)	1.87 (1.57–2.22) ***	0.80 (0.59–1.08)	0.34 (0.22–0.53) ***	1
Fibrates only	**≤360 DDD**	**361–720 DDD**	**≥721 DDD**	**Study group II**
	**N = 1132**	**N = 224**	**N = 136**	**N = 15,482**
AMD cases (n, %)	33 (2.91)	7 (3.12)	7 (5.14)	268 (1.73)
Crude HR (95% CI)	1.70 (1.18–2.44) **	1.83 (0.86–3.87)	3.05 (1.44–6.47) ***	1
Adjusted HR (95% CI)	1.60 (1.11–2.31) *	1.77 (0.83–3.76)	3.43 (1.61–7.30) **	1
Statins only	**≤360 DDD**	**361–720 DDD**	**≥721 DDD**	**Study group II**
	**N = 6791**	**N = 2327**	**N = 1246**	**N = 15,482**
AMD cases (n, %)	185 (2.72)	32 (1.37)	11 (0.88)	268 (1.73)
Crude HR (95% CI)	1.58 (1.31–1.91) ***	0.79 (0.55–1.14)	0.51 (0.28–0.93) *	1
Adjusted HR (95% CI)	1.49 (1.24–1.81) ***	0.82 (0.57–1.19)	0.56 (0.31–1.03)	1

***: *p* < 0.001; **: *p* < 0.01; *: *p* < 0.05; HR, hazard ratio; AMD, age-related macular degeneration; DDD, defined daily dose; study group I, patients with hyperlipidemia with the use of statins and/or fibrates. HR adjusted for age, gender, hypertension, diabetes mellitus, cerebrovascular disease, coronary artery disease, heart failure, atrial fibrillation, myocardial infarction, atherosclerosis, stroke, peripheral vascular disease, glaucoma, diabetic retinopathy, obesity, tobacco use disorder & alcohol abuse, cataract surgery, aspirin, warfarin, hormone replacement therapy, nonsteroidal anti-inflammatory drug, and antihypertensive agents.

## Data Availability

We must follow the Computer-Processed Personal Data Protection Law and related regulations in Taiwan. The datasets for this manuscript are not publicly available because of the HWD protection policy.

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
