# Peer review of "Association between Antihyperlipidemic Agent Use and Age-Related Macular Degeneration in Patients with Hyperlipidemia: A Population-Based Retrospective Cohort Study"

_biomedicines, 2023, doi:10.3390/biomedicines11061508_

Round 1
Reviewer 1 Report (Previous Reviewer 1)
The paper is fine and the author appropriately answered to all the issues I raised.
Reviewer 2 Report (Previous Reviewer 2)
The authors have answered all my queries from the previous review.
No further questions.
Nothing to comment.
Reviewer 3 Report (Previous Reviewer 3)
Dear Editor,
I carefully read the revised version of the manuscript by Chen et al. that is significantly improved in comparison with the original version. All my previous concerns have been addressed by the authors, so that I warmly suggest to publish the article in the Journal.
This manuscript is a resubmission of an earlier submission. The following is a list of the peer review reports and author responses from that submission.
Round 1
Reviewer 1 Report
I read with great interest the paper “Association between antihyperlipidemic agent use and age-related macular degeneration in patients with hyperlipidemia. A population-based retrospective cohort study” by Chen et al.
Paper design is fine. English should bi moderately revised. There are figures and tables of good qualities.
Comments:
1. Page 1 Line 45: please revise is indicated with has reported
2. Page 2 line 1: I suggest that Shihpai and Taipei are the two Taiwan cities where has been performed the study. Taiwan is unnecessary. Please revise. Same line 2-3,
3. Limitation: the population is matched for age and gender, though it is not matched for other comorbidities and laboratory parameters. The populations compared are not homogeneous, thus increasing the bias. Please discuss it and report in the appropriate section.
4. Introduction: high cholesterol levels have been associated to several other eyes disease such retinopathy, and has been also suggested in AMD (doi: 10.1016/j.diabres.2019.03.028). I suggest to improve the introduction by adding this sentence.
Reviewer 2 Report
I think that this is a nice work showing the clear association between hypolipemic agents and age-related macular degeneration in patients with hyperlipidemia.
The paper is generally well-written and all required ethical disclosures are presented.
1. I would suggest authors change subtitles in the tables to make them clearer - for example: replace Study group II with "patients with hyperlipidemia not receiving statins/fibrates", etc. In a current way, it is presented making it difficult for the reader to follow which group represents what.
2. I think it would be beneficial to show the effect of certain subclasses of drugs on the time-to-event (in this case AMD), for example, to make stratification with respect if the statins vs. fibrates used.
3. Equally so, it would be nice to show the effect of high-dose statins vs. lower-dose statins on these outcomes. I think these subanalyses would be beneficial and would better inform the clinician.
4. Did the author's record any side-effect events of statin used? I think this would be nice to show from the standpoint of the safety-to-efficacy ratio.
Reviewer 3 Report
My further comments and suggestions for the authors are the following:
- In their manuscript, the authors refer to "antihyperlipidemic agents". However, I think that it would be more accurate if they designed and performed two different analyses by type of LLT (e.g. LDL-C lowering drugs and, separately, fibrates and PUFAs).
- Page 4, Line 4: The authors should specify how the normal distribution of the variables was assessed.
- Page 4, Line 5: The authors should specify if they performed the Levene's test before the Student's T test.
- Page 4, Line 8: The authors should assess also the median survival time in the groups.
- The limitations of the study should be further and more deeply discussed in the Discussion.
- As regards the Confounders (Page 3, Lines 41 and ahead), it is not clear to me why the authors considered either presence of hypertension and treatment with antihypertensive drugs. In effect, patients assuming BP-lowering drugs always have a diagnosis of hypertension. Then, the covariate "treatment with antihypertensive drugs" should be obsolete.
Round 2
Reviewer 2 Report
The authors were not able to address most of my questions because they did not originally perform such analysis or have captured the data regarding it.
Reviewer 3 Report
Dear Editor,
Unfortunately this study is still inconsistent.